# Silencing of Putative Plasmodesmata-Associated Genes *PDLP* and *SRC2* Reveals Their Differential Involvement during Plant Infection with Cucumber Mosaic Virus

**DOI:** 10.3390/plants14030495

**Published:** 2025-02-06

**Authors:** Richita Saikia, Athanasios Kaldis, Carl Jonas Spetz, Basanta Kumar Borah, Andreas Voloudakis

**Affiliations:** 1Laboratory of Plant Breeding and Biometry, Faculty of Crop Science, Agricultural University of Athens, 11855 Athens, Greece; richasaikia2195@gmail.com (R.S.); akaldis2003@hotmail.com (A.K.); 2Department of Agricultural Biotechnology, Assam Agricultural University, Jorhat 785013, Assam, India; basantabora@gmail.com; 3Division of Biotechnology and Plant Health, Norwegian Institute of Bioeconomy Research, 1433 Ås, Norway; carl.spetz@nibio.no

**Keywords:** tobacco rattle virus (TRV)-based gene silencing, plant–virus interaction, disease resistance, resistance and susceptibility gene, solanaceous plants

## Abstract

Plant viruses utilize a subset of host plasmodesmata-associated proteins to establish infection in plants. In the present study, we aimed to understand the role of two plant genes, one encoding a putative plasmodesma located protein (PDLP) and a homolog of soybean gene regulated by cold 2 protein (SRC2) during Cucumber mosaic virus (CMV) infection. Virus-induced gene silencing (VIGS) was used to silence *PDLP* and *SRC2* genes in *Nicotiana benthamiana* and in two related solanaceous plants, *N. tabacum* and *Capsicum chinense* Jacq. (Bhut Jolokia). Up to 50% downregulation in the expression of the *PDLP* gene using the TRV2-*PDLP* VIGS construct was observed in *N. benthamiana* and *N. tabacum* while, using the same gene construct, 30% downregulation of the target mRNA was observed in *C. chinense*. Similarly, using the TRV2-*SRC2* VIGS construct, a 60% downregulation of the *SRC2* mRNA was observed in *N. benthamiana*, *N. tabacum*, and a 40% downregulation in *C. chinense* as confirmed by qRT-PCR analysis. Downregulation of the *PDLP* gene in *N. benthamiana* resulted in delayed symptom appearance up to 7–12 days post inoculation with reduced CMV accumulation compared to the control plants expressing TRV2-*eGFP*. In contrast, *SRC2*-silenced plants showed enhanced susceptibility to CMV infection compared to the control plants. Our data suggest that the *PDLP* gene might facilitate infection of CMV, thus being a susceptibility factor, while the *SRC2* gene could play a role in resistance to CMV infection in *N. benthamiana*.

## 1. Introduction

To establish an infection, most plant viruses utilize specialized membrane-linked cytoplasmic channels known as plasmodesmata (PD) for intercellular trafficking of virus particles in the host [1]. They encode at least one dedicated protein termed movement protein (MP) that can interact with host proteins and modify the structure of PD [2,3]. Viral MPs can form tubules that assemble in PD and thereby manipulate the size exclusion limit of PD for the transport of viral complexes to the adjacent cells [4]. Viruses encoding such tubule-forming MPs belong to six major families: Badnaviridae, Bromoviridae, Caulimoviridae, Ilaviridae, Secoviridae, and Tospoviridae [5,6]. To date, only a few of the host proteins localized to PD have been shown to interact with viral MPs. These include proteins such as calreticulin [7], protein kinase PAPK1 [8], remorin [9], endoplasmic reticulum (ER)-embedded proteins including reticulons [10,11], plasmodesmata-associated class 1 reversibly glycosylated polypeptide [12], ankyrin-repeat containing protein (ANK) [13], synaptotagmin [14], and other plasmodesmata-located proteins (PDLPs) [5,15]. It is suggested that ANK together with β-glucanases could regulate the size exclusion limit of plasmodesmata [13]. ER-embedded proteins have been shown as positive regulators for viral movement and spread of cucumber mosaic virus (CMV) [10]. PDLPs have been shown as positive regulators for viral movement and spread of grapevine fan leaf virus (GFLV) and cauliflower mosaic virus (CaMV) [5]. Another viral protein of CaMV, named P6, colocalized with host PDLP1 and CaMV MP in the PD [16]. The study also showed that P6 physically interacts with a C2 calcium-dependent membrane-targeting protein, SRC2 (Soybean Response to Cold) from Arabidopsis, and this protein is capable of a close association with the tubule formation in the PD. Although the expression of pepper *CaSRC2-1* gene was shown to be upregulated during bacterial and viral infection in chilli (*Capsicum annum*) [17], little is known about the exact role of *SRC2* and *PDLP* genes during plant viral infection, particularly for broad host range viruses like CMV. CMV infects more than 1000 plant species [18,19]. These include some of the most valuable horticultural crops such as chilli, cucumber, okra, squash, and tomato [20]. CMV has a linear tripartite genome of positive-sense single-stranded (ss) RNA (RNA1/RNA2/RNA3) and belongs to the family Bromoviridae [21]. It is also known that the MP of CMV localizes to the PD like other viruses belonging to Bromoviridae family [22]; however, the exact role of host proteins during CMV infection is not well defined as of yet.

RNA interference (RNAi) or post-transcriptional gene silencing (PTGS) is a natural surveillance system adopted by plants that can regulate levels of endogenous or foreign transcripts by degrading them in a sequence-specific manner. The PTGS system becomes activated in response to the formation of double-stranded RNA (dsRNA), which involves sequence-specific recognition of target mRNA molecules based on sequence homology and subsequent degradation of these molecules by host ribonucleases and other proteins in the RNA induced silencing complex (RISC), thereby inhibiting their translation process [23]. The RNAi mechanism can be evoked using recombinant viral vectors that can efficiently deliver silencing inducer molecules to plant cell and study target gene function. This reverse genetic approach of studying plant genes using viral vectors is called virus-induced gene silencing (VIGS). Some of the recombinant viral vectors are derived from the genome of Potato virus X (PVX), Tobacco mosaic virus (TMV), and Tobacco rattle virus (TRV), of which TRV- and PVX- have been widely utilized [24,25]. The TRV-VIGS has been used to characterize genes in many plant species related to plant developmental processes, biotic and abiotic stress tolerance, symbiosis, metabolite synthesis, and plant evolution [25,26,27,28]. TRV-VIGS was first reported in *N. benthamiana* for silencing an endogenous gene, namely *phytoene desaturase* (*PDS*), and this vector system was further modified [29,30,31]. Since then, several disease resistance and susceptibility genes were identified against viral infection in plant species using the TRV-VIGS system [32,33,34].

In the present study, the role of two genes, one encoding putative plasmodesmata-localized protein PDLP and a protein homolog of SRC2 from *Nicotiana benthamiana* and *N. tabacum* were studied using the TRV-based VIGS system for their involvement in plant defense against or susceptibility to CMV.

## 2. Results

### 2.1. Homology Analysis of PDLP and SRC2 Genes in the Nicotiana and Capsicum Species

Using BLASTn and phylogenetic analysis with known *PDLP* genes from *Arabidopsis thaliana*, we selected a homolog of *PDLP* gene (XM_016610375.1) from *N. tabacum* which was identified in *N. benthamiana* (Niben101Scf03456g00018.1, 95.16% sequence identity with 99% query cover), *C. annum* (XM_016721975.2, 80.26% sequence identity with query cover 99%) and *C. chinensis* (MCIT02000006.1, 75.26% sequence identity with query cover 100%) (Figure 1A, Appendix A). Similarly, a homolog of the *SRC2* gene (XM_016630487.1) from *N. tabacum* was selected which was identified in *N. benthamiana* (DQ465395.1, 93.26% sequence identity with 99% query cover), *C. annum* (NM_001324570.1, 76.98% sequence identity with 100% query cover), and *C. chinensis* (AB442165.1, 76.68% sequence identity with query cover 94%) (Figure 1B, Appendix A) by using BLASTn. The homologous sequences of *PDLP* and *SRC2* were used to design primers for PCR analyses (RT-PCR and RT-qPCR) and for construction of the VIGS clones.

### 2.2. Relative Expression of PDLP and SRC2 Genes During CMV Infection in N. benthamiana

To investigate the expression pattern of *PDLP* and *SRC2* genes, *N. benthamiana* leaves were inoculated with CMV, phenotyped, and sampled at 2, 8, and 15 days post inoculation (dpi). Typical downward leaf curling and mosaic symptoms were observed at 8 dpi, but the symptoms were more prominent at 14 dpi (Figure 2). Using semi-quantitative RT-PCR, the CMV *CP* was detected only at 8 and 15 dpi in the inoculated leaves, but not in the 2 dpi leaf samples (Figure 3).

No CMV-specific band was detected in the systemic leaves of the plants treated with the phosphate buffer (mock) (Figure 3). The expression levels of the *PDLP* and *SRC2* genes were observed, and it was found that the *PDLP* gene expression was nearly the same between the mock and CMV inoculated plants, suggesting that the gene was constitutively expressed irrespective of the CMV infection. However, the expression of the *SRC2* gene was upregulated upon CMV inoculation, as observed at 8 and 15 dpi compared to the mock (Figure 3). Therefore, to understand the functional role of these two genes during viral infection, we decided to silence the genes in *N. benthamiana* employing a TRV-based VIGS strategy.

### 2.3. VIGS of PDLP and SRC2 Genes in Tobacco and Bhut Jolokia Pepper

#### 2.3.1. Designing of VIGS Constructs and Silencing of *PDLP* and *SRC2* Genes in *N. benthamiana*, *N. tabacum*, and *C. chinense*

A conserved region of *PDLP* (383 bp) and *SRC2* (364 bp) genes from tobacco, *N. benthamiana*, *C. annum*, and *C. chinense* was identified using multiple sequence alignment by Clustal Omega (Appendix A). The conserved regions of *PDLP* and *SRC2* gene sequences were amplified using PCR with the respective primer pairs (Table 1) and cloned into the pTRV2 plasmid vector. Colony PCR of the transformed *E. coli* cells revealed positive bacterial colonies harboring the cloned constructs, pTRV2-*NtPDLP* and pTRV2-*NtSRC2*. Sanger sequencing of at least three positive clones from each gene fragment confirmed the presence of *NtPDLP* and *NtSRC2* gene fragments in pTRV2 in sense orientation. Alignment of the sequenced *PDLP* and *SRC2* gene fragments with *N. tabacum PDLP* (XM_016610375.1) and *SRC2* (XM_016630487.1) revealed 100% nucleotide identity.

Transient expression of *PDLP* and *SRC2* gene fragments cloned in the pTRV2 expression vector was carried out through agroinfiltration in *N. benthamiana*, *N. tabacum*, and *C. chinense* leaves (Appendix A). The phytoene desaturase gene (*PDS*) from *N. benthamiana* cloned in pTRV2, which results in photobleaching effect upon gene silencing, was used as a positive control (Appendix A). The silencing experiment for *PDLP* and the *SRC2* gene in *N. benthamiana* was repeated two times, while the silencing experiment for these genes in *N. tabacum* and *C. chinense* plants was performed once. The silencing efficiency of the target genes (*PDLP* and *SRC2*) was evaluated using RT-qPCR and compared between the silenced and the control plants. In the *PDLP*-silenced group, the expression level of the *PDLP* gene was 0.4- to 0.6-fold lesser than the GFP control with the highest downregulation at 9 dpa (Figure 4). Similarly, the expression of the *SRC2* gene in the *SRC2*-silenced group was 0.3- to 0.5-fold lesser than the GFP control with highest downregulation observed at 21 dpa (Figure 4).

The silencing of *PDLP* and *SRC2* genes in *N. tabacum* and *C. chinense* Bhut Jolokia plants was analyzed at 14 dpa (Figure 5). Using the same VIGS constructs, a considerable level of *PDLP* and *SRC2* gene silencing was also observed in all species tested, despite having sequence polymorphisms among these species. The expression of the *PDLP* gene in *N. tabacum* and *C. chinense* was 0.5-fold and 0.7-fold lesser than the GFP control (a reduction in gene expression by 50% and 30%, respectively) (Figure 5). Similarly, in both *N. tabacum* and *C. chinense* Jacq., the expression of the *SRC2* gene was downregulated to approximately 0.4-fold and 0.6-fold, respectively, in comparison to the GFP control (downregulation up to 60% and 40% in *N. tabacum* and *C. chinense* Jacq., respectively) (Figure 5).

#### 2.3.2. Disease Development in *PDLP*- and *SRC2*-Silenced *N. benthamiana* Plants Upon CMV Infection

The *PDLP-* and *SRC2*-silenced *N. benthamiana* plants were visually observed upon CMV infection and disease symptoms were recorded at different time points, i.e., 7, 12, and 19 dpi (Figure 6 and Appendix A). In the *eGFP* control plants, the typical mosaic symptom of CMV was observed at 7 dpi and the symptoms gradually intensified with severe downward leaf curling at 19 and 21 dpi. In contrast, the *PDLP*-silenced plants displayed reduced viral symptoms in comparison to the *eGFP* control. In the *SRC2*-silenced plants, leaf curling similar to or even greater than the *eGFP* control plants was observed at 19 and 21 dpi, indicating that silencing of *SRC2* might have promoted CMV infection in *N. benthamiana* (Figure 6 and Appendix A).

#### 2.3.3. Relative Quantification of CMV Titer in *PDLP*- and *SRC2*-Silenced *N. benthamiana* Plants

Systemic (uninoculated) leaves were collected at different time points upon CMV infection, and the viral titer was evaluated using RT-qPCR analyses. Significant reduction in the CMV titer was observed at 7 and 12 dpi but not at 19 dpi in the *PDLP*-silenced plants in comparison to the control plants (i.e., eGFP) by RT-qPCR analysis (Figure 7). Similarly, it was found that the level of CMV was significantly higher in the *SRC2*-silenced plants when compared to the eGFP control plants at 12 and 19 dpi by using RT-qPCR (Figure 7).

#### 2.3.4. Callose Deposition in the *PDLP*- and *SRC2*-Silenced *N. benthamiana* Plants Upon CMV Infection

To strengthen the RT-PCR and RT-qPCR results, a callose deposition assay was conducted. Callose deposits initiated at 7 dpi and progressively increased, with higher callose deposits observed at 19 dpi in the eGFP control plants after CMV infection. In contrast, lesser callose deposits were observed in the *PDLP*-silenced plants at 7, 12, and 19 dpi. In the *SRC2*-silenced plants, callose deposition exhibited a near-equivalent level to that of the control group at 7, 12, and 19 dpi (Appendix A).

#### 2.3.5. Accumulation of Reactive Oxygen Species (ROS) in the *PDLP*- and *SRC2*-Silenced *N. benthamiana* Plants Upon CMV Infection

The level of hydrogen peroxide (H_2_O_2_), a ROS molecule, was assessed in the *PDLP-* and *SRC2*-silenced plants following CMV infection. Remarkably, elevated levels of H_2_O_2_ were detected in systemic leaves of the control and *SRC2*-silenced groups at 7, 12, and 19 dpi (Appendix A). In contrast, the concentrations of H_2_O_2_ in *PDLP*-silenced plants were notably lower in comparison to the control and *SRC2*-silenced groups upon CMV infection at 7, 12, and 19 dpi (Appendix A).

## 3. Discussion

It is known that plasmodesmata-associated proteins and plasmodesmata (PD) are crucial for cell-to-cell movement of plant viruses [35]. *PDLP*s and *SRC2* are plasmodesmata-associated genes [5,16,35]. Some viruses recruit movement proteins that localize to the PD and interact with the PDLP protein to form a movement tubule facilitating, as a result, the transport of viral particles allowing systemic spread of the virus in the plant. The movement protein of CMV also localizes in the plasmodesmata [22]; however, the role of PDLPs in viral disease development is not well studied. The SRC2 protein is also a PD-associated membrane protein that colocalizes with *Arabidopsis* AtPDLP and interacts with the movement protein of cauliflower mosaic virus (CaMV) *in planta* [16]. The expression of pepper *CaSRC2-1* gene was upregulated during bacterial and viral infection in *Capsicum annum* [17]. Little is known about the expression of the *SRC2* gene of *Nicotiana* spp. during infection with CMV. Thus, these two genes, *PDLP* and *SRC2*, are promising candidates for promoting disease susceptibility or resistance to CMV in tobacco; therefore, they were functionally characterized in our study.

The expression of these two tobacco genes was transiently silenced utilizing a TRV-based VIGS strategy. The expression of *PDLP* upon CMV infection in *N. benthamiana* plants was consistent between mock and CMV inoculated samples, suggesting that the expression of *PDLP* is not significantly affected by CMV infection. On the other hand, the expression of *SRC2* was slightly upregulated at 8 dpi upon CMV infection, corresponding with the initiation of CMV expression at the same time point (Figure 3). Similar observations were made in pepper (*C. annum*) and soybean (*Glycine max*) [17,36]. Takahashi and Shimosaka (1997) found that *SRC2* expression in soybean was upregulated in response to cold stress and mechanical wounding [36]. Additionally, Kim et al. (2008) observed an induction of *SRC2* gene expression in pepper during bacterial and viral infection [17].

Through TRV-based VIGS, an average of about 80% downregulation can be achieved in the endogenous transcript levels of the targeted gene compared to vector control plants [25]. The initiation of silencing generally starts within 2–3 weeks after agroinfiltration [25]. The efficiency of silencing may vary due to various factors such as the VIGS-vector infection, multiplication, and systemic spread, the level of viral-vector titer in the host, the homology between the target gene fragment in the viral-vector and the endogenous target gene mRNA, and the environmental conditions during plant growth [25]. The present study revealed significant downregulation in both *PDLP* and *SRC2* at 4, 9, and 21 dpa (Figure 4). RT-qPCR analysis revealed a significant downregulation of the *PDLP* gene by 50% in *N. tabacum*, *N. benthamiana*, and 30% in *C. chinense* (Figure 4 and Figure 5), along with the downregulation of the *SRC2* gene by 60% in *N. tabacum*, *N. benthamiana*, and 40% in *C. chinense* (Figure 3 and Figure 4). Similar observations were reported in the knockdown lines of *PDLP-1*, *PDLP-2*, and *PDLP*-3 triple mutants in *Arabidopsis thaliana* which was resistant to powdery mildew infection [37]. Kim et al. (2008) proposed that *SRC2* from pepper (*C. annuum*) plays an essential role in inducing resistance to bacterial and viral pathogens by eliciting the hypersensitive response (HR) to restrict pathogen growth [17]. Liu et al. (2016) provided evidence that SRC2 recognizes INF1, an elicitor protein of *Phytophthora capsici*, and elicits HR in pepper [38]. They showed that silencing of the *SRC2* gene increased susceptibility of pepper plants to *P. capsici* infection. In addition, the silencing experiment did not reveal any developmental changes such as alteration in plant height in the *PDLP-* and *SRC2*-silenced plants when compared to the control plants (TRV2-*eGFP*). It is noteworthy that this is the first report of a TRV-based silencing of plant endogenous genes in *C. chinense* Bhut Jolokia. No developmental alterations were observed in *N. benthamiana* and *C. chinense* in any VIGS experiment, except for *N. tabacum* where development changes such as growth inhibition and minor leaf crumbling were observed at 14 dpa (Appendix A). Photobleaching, indicative of *PDS* gene silencing, started at 9 dpa and became even more prominent at 14 and 21 dpa (Appendix A), suggesting that silencing effect commenced approximately 1–2 weeks post agroinfiltration. This experiment allowed us to determine the silencing effect of plant endogenous genes with the TRV-based VIGS strategy, which was demonstrated within 14 dpa.

The current investigation unveiled that silencing of the *PDLP* in *N. benthamiana* resulted in a notable reduction in the accumulation of the CMV titer in the systemic leaves. The *PDLP* gene family comprises a set of multiple genes, specifically *PDLP1* to *PDLP8* in Arabidopsis, with predicted molecular weights ranging from 30 to 35 kDa [39]. The downregulation of the *PDLP* gene resulted in delayed symptoms of CMV infection up to 12 dpi but not until 19 dpi, suggesting that targeting multiple *PDLP* genes, if present, of *N. benthamiana* could potentially lead to further delay in viral symptom development. Another possibility is that CMV might exploit other plasmodesmata proteins to facilitate transport at later stages of systemic infection. Therefore, it is important to understand the role of other PDLPs in *N. benthamiana*. It can be speculated that the reduced H_2_O_2_ accumulation observed in the *PDLP*-silenced plants (Appendix A) is likely a result of decreased CMV accumulation compared to the *TRV2-eGFP*-treated plants. It is known that in a CMV–plant compatible interaction, there is an increase in H_2_O_2_ accumulation [40] that agrees with our observations. The reduced leaf mottling and mosaic symptoms in the *PDLP*-silenced plants correlate nicely with the reduced CMV titer and ROS accumulation (Figure 7 and Appendix A). Plants produce callose deposits in the cell periphery as a defense mechanism in response to pathogen invasion. Previous studies have demonstrated that overexpression of *PDLP1* and *PDLP5* enhances callose deposition at PD [37,41], which is consistent with the low callose deposition in our *PDLP*-silenced plants. Lim et al. (2016) demonstrated that the overexpression of *PDLP5* resulted in a reduction in the trafficking of azelaic acid (AzA) and glycerol-3-phosphate (G3P) through PD [42]. This impairment of PD trafficking was associated with a significant decline in Systemic Acquired Resistance (SAR) in plants. Similarly, Carella et al. (2015) observed that overexpressing of either *PDLP1* or *PDLP5* in Arabidopsis plants led to the loss of systemic movement of the crucial SAR protein, DEFECTIVE IN INDUCED RESISTANCE1 (DIR1), resulting in a compromised SAR [43]. The overexpression of *PDLP*s in *A. thaliana* correlates with an enhanced systemic spread of pathogens suggesting their role in promoting pathogen spread. We could hypothesize that *PDLP* silencing would augment SAR for the benefit of the plant and having negative effect for CMV. In addition, PDLPs are known to be involved in viral movement [5,6,15,16]. For example, for tubule-guided viral movement through plasmodesmata, the viral movement proteins (MPs) of GFLV and CaMV in Arabidopsis [5] and cowpea mosaic virus in *N. benthamiana* [15] interact with the host PDLPs. The interaction of viral MPs with host PDLPs results in the formation of movement tubules that allow cell-to-cell movement of plant viruses across adjacent cells [5]. CMV-MP also localizes in the plasmodesmata as it is a member of the ‘30K MP superfamily’ group [22]. This finding is consistent with the results obtained from our *PDLP*-silenced *N. benthamiana* plants, where a reduction in *PDLP* expression led to decreased CMV accumulation. While the exact function of PDLPs in virus movement is still under investigation, future research can be prioritized to understand the structural interactions between tubule-forming MPs (e.g., CMV) and PDLPs. Overall, these findings suggest that *PDLP* silencing of *N. benthamiana* led to decreased CMV accumulation in systemic tissue, suggesting that PDLPs are possibly involved in facilitating the systemic spread of CMV and as a result increase the viral titer, so they should be considered as susceptibility factors. Generating knockout/knockdown lines of *PDLP* genes in Bhut Jolokia and *N. benthamiana* using the CRISPR/Cas9 system could aid in developing virus-resistant plants against CMV.

In contrast, silencing of *SRC2* resulted in increased susceptibility to CMV infection (Figure 7), indicating that SRC2 might be involved in the resistance to CMV in *N. benthamiana*. Kim et al. 2008 [17] reported that during both bacterial and viral infections in *C. annum*, the pepper encoding *CaSRC2*-1 (a single C2 domain-containing protein) gene was upregulated. Limited information exists regarding the expression patterns of the *SRC2* gene in *Nicotiana* spp. during the infection with CMV (our data indicate a slight *SRC2* upregulation upon CMV infection [Figure 3]). A recent study found that the silencing of *GmSRC2* in soya resulted in a significant enhancement of disease symptoms and an increase in the biomass of *Phytophthora sojae* [44]. A structural characteristic identified in AtSRC2.2 is its C2 domain, which serves as a lipid-binding domain found in numerous eukaryotic proteins [16]. The C2 domain, first identified in Ca^2+^-dependent isoforms of protein kinase C, is integral to cellular signalling in plants and animals [17]. Plant proteins harboring C2 domains have been associated with facilitating the intercellular spread (cell-to-cell movement) of plant viruses [45]. An interesting characteristic of AtSRC2.2 is that its C2 domain shares 36% similarity with AtSYT4, a member of the synaptotagmin family, possessing two C2 domains (C_2_A and C_2_B) [16]. Not much is known about the involvement of AtSYT4 in virus movement, but there is another synaptotagmin, AtSYTA, which is involved in the movement of several viruses, such as cabbage leaf curl virus, tobacco mosaic virus, and squash leaf curl virus by interacting with their respective MPs [45]. It is noteworthy that AtSYTA is also associated with regulating freezing tolerance [46], a trait shared among SRC2 proteins. Several studies have suggested that the entry of Ca^2+^ is a key factor in initiating pattern-triggered immunity (PTI) in plant cells [44]. In the *SRC2*-silenced plants, a higher H_2_O_2_ accumulation was observed (Appendix A) that could be the consequence of higher CMV load (Figure 7) in the compatible interaction with *N. benthamiana* agreeing with [40]. Therefore, further investigation is needed to understand the role of *SRC2* in plant defense response. The notion that enhancing resistance to CMV in plants could be achieved by overexpressing the *SRC2* gene is now a hypothesis to test.

## 4. Materials and Methods

### 4.1. Plant Materials and Growth Conditions 

*Nicotiana benthamiana*, *N. tabacum*, and *Capsicum chinense* Jacq. (Bhut Jolokia) plants were grown in a greenhouse operating at 20 ± 3 °C and 70% humidity with a 16 h/8 h day/night cycle. Three- to four-week-old plants were selected for performing the VIGS experiment, and they were divided into four groups—TRV1 + TRV2-*NtPDLP*, TRV1 + TRV2-*NtSRC2*, TRV1 + TRV2-*eGFP* (negative control), and TRV1 + TRV2-*NbPDS* (positive control for silencing).

### 4.2. Development of TRV-Based Recombinant Constructs for VIGS 

The TRV-based VIGS system, developed by Liu et al., 2002 [31], was used to generate the recombinant VIGS constructs for silencing of putative *PDLP* and *SRC2* genes in Solanaceous plants including *Nicotiana benthamiana*, *N. tabacum*, and *C*. *chinense.* TRV is a bipartite ss (+) RNA virus comprising two genomes, RNA1 and RNA2. Both TRV-RNA1 and TRV-RNA2 genomes were previously cloned in the binary plasmid vectors pBIN19 and pCambia0390, respectively [31]. The pTRV-RNA2 plasmid vector was digested by the *Bam*HI-HF restriction enzyme (New England Biolabs, USA) to obtain a linearized plasmid DNA. The digested product was subjected to gel electrophoresis for confirmation along with the quantification of the digested pTRV2 and was stored at −20 °C for further use.

For the integration of the *PDLP* and *SRC2* gene fragments, the In-Fusion^®^ HD seamless cloning system (Takara Bio Inc., Shiga, Japan) was employed. The nucleotide sequences corresponding to the putative *PDLP* (XM_016610375.1) and *SRC2* (XM_016630487.1) genes of *N*. *tabacum* were employed as queries in the VIGS tool provided by the Sol Genomics Network (SGN) to identify the optimal fragment for gene silencing purposes. Primers were designed for the amplification of the fragments of *PDLP* and *SRC2* gene using the primer3 tool (http://primer3.ut.ee/) (accessed on 15 February 2022). The designed primer had a 15-base extension at the 5′ end homologous to the linearized pTRV2 vector (Table 1). The amplification of *PDLP* and *SRC2* gene fragments was performed by RT-PCR utilizing RNA from *N. tabacum* leaves as a template. The PCR-amplified product underwent confirmation and quantification through 1.5% agarose gel electrophoresis and was subsequently cloned into the linearized pTRV2 vector as per the manufacturer’s instructions (In-Fusion^®^ HD cloning kit, Clontech, Mountain View, CA, USA). The recombinant plasmid constructs (pTRV2-*NtPDLP* and pTRV2-*NtSRC2*) were transformed into *E. coli* Stellar Competent Cells (Takara Bio Inc., Shiga, Japan), followed by a heat shock at 42 °C for 30 s. The bacterial cultures were spread over Luria–Bertani (LB) plates containing kanamycin (50 µg/mL) and were subjected to overnight incubation at 37 °C for colony development. Colony PCR was carried out to confirm the presence of recombinant plasmids in the transformed bacterial colonies, using specific primers (TRV2-1530F and TRV2-1809R) that bind on the TRV2 plasmid vector (Table 1). Two positive transformants for each of the recombinant plasmids were selected for plasmid isolation using NucleoSpin^®^ Plasmid Kit (Machery-Nagel, Düren, Germany) and stored for further use. The confirmation of the successful integration of the *NtPDLP* and *NtSRC2* gene fragment into the pTRV2 vector was validated through Sanger sequencing (Azenta Life Sciences UK, Essex, UK).

**Table 1 plants-14-00495-t001:** List of primers used in this study.

Name of Primer	Sequence (5′ to 3′) *	Product Size (bp)	Target Species	Function
Nt_PDLP-INF-378F Nt_PDLP-INF-760R	GCCTCCATGGGGATCGTGTACAAAGGCTGTGCTAA GCTCGGTACCGGATCGTCTCAAGCCTATCACTAAAC	413	*N. tabacum* *N. benthamiana*	Cloning of *PDLP* gene fragment to pTRV2
Nt_SRC2-INF-125F Nt_SRC2-INF-488R	GCCTCCATGGGGATCCATTAGATATCAAAGTTATTGC GCTCGGTACCGGATCGGTTTTCCGCTCTCTGTTAT	394	*N. tabacum* *N. benthamiana*	Cloning of *SRC2* gene fragment to pTRV2
eGFP-INF-286F eGFP-INF-646R	GCCTCCATGGGGATCGAGCGCACCATCTTCTTCAA GCTCGGTACCGGATCGCTTCT CGTTGGGGTCTTTG	391	**-**	Cloning of *eGFP* gene fragment to pTRV2
TRV2-1530F TRV2-1809R	GTTTTTATGTTCAGGCGGTTC TCAAGATCAGTCGAGAAT GTCA	280	*E. coli* *A. tumefaciens* agroinfiltrated plant species	Colony PCR, Detection of TRV2
TRV1-257F TRV1-491R	GCTGAGCAGAGGAGTCATTTC ACCCATGAACCATGTTTTTGT	235	*E. coli* *A. tumefaciens* agroinfiltrated plant species	Detection of TRV1
Nb_PDLP-RT-F Nb_PDLP-RT-R	GGGACTGTGTGAACTGTGTG ACTCCTAACCCCACACCAAC	210	*N. tabacum* *N. benthamiana*	Gene expression analysis for *PDLP*
Nb_SRC2-RT-F Nb_SRC2-RT-R	ATCCACCCGTACAACAACCT AGCAGCCATCTCTCCAACAT	164	*N. tabacum* *N. benthamiana*	Gene expression analysis for *SRC2*
CMV_CP_RT-F CMV_CP_RT-R	GAAGCTTGTTTCGCGCATTC TCCGGATGCTGCATACTGAT	175	CMV	Gene expression analysis for CMV *CP*
Nb_EF1a-F Nb_EF1a-R	AGCTTTACCTCCCAAGTCATC AGAACGCCTGTCAATCTTGG	116	*N. tabacum* *N. benthamiana*	Gene expression analysis for *EF1a* [47]

***** The region that anneals to the TRV2 vector for the In-Fusion cloning is underlined.

To serve as a negative control in the VIGS assay, we introduced a 361 nt segment from the enhanced GFP (*eGFP*) gene (a gene that does not exist in plants) into the pTRV2 plasmid vector. The insert fragment was derived through PCR amplification using In-Fusion primers designed for this purpose, with a pBIN61-eGFP plasmid vector employed as the template. All subsequent steps for constructing the engineered pTRV2_eGFP plasmid were executed by the aforementioned procedures. The positive control in the VIGS assay involved the targeted silencing of a fragment of *N. benthamiana* phytoene desaturase gene (PDS) exhibiting a photobleaching phenotype. This control was employed to validate the efficiency of the gene silencing system.

### 4.3. Transformation and Mobilization of VIGS Constructs into Agrobacterium Tumefaciens (GV3101 Strain)

The recombinant plasmids (pTRV2-*NtPDLP* and pTRV2-*NtSRC2*) were transformed into *Agrobacterium tumefaciens* competent cells (GV3101 strain) using the freeze–thaw method with minor modifications [48]. The plasmids were added to the surface of frozen GV3101 competent cells and promptly placed in a heat block at 37 °C for 5 min. Following the heat treatment, the transformed GV3101 competent cells were mixed with LB and subjected to shaking at 28 °C for 4 h. The transformed bacterial cells were spread onto an LB agar plate containing 50 µg/mL kanamycin and 25 µg/mL rifampicin and incubated at 28 °C for 2–4 days for colony development. Positive colonies were further confirmed by colony PCR as mentioned earlier for *E. coli*. Similarly, the pTRV1 binary plasmid vector carrying the TRV-RNA1 genome was introduced into GV3101 competent cells.

### 4.4. Agrobacterium-Mediated Transient Expression and Virus Induced Gene Silencing

VIGS constructs were delivered to three- to four-week-old *N. benthamiana*, *N. tabacum*, and *C. chinense* plants via agroinfiltration. Primary and secondary cultures of the relevant Agrobacteria were cultured in LB supplemented with kanamycin (50 μg/mL) and rifampicin (25 µg/mL) at 28 °C in a shaker incubator overnight at 220 rpm. The secondary cultures were centrifuged at 3000× *g* for 10 min. The supernatant was discarded, and the resulting cell pellet was resuspended in a resuspension buffer (10 mM 2-(N-Morpholino) ethanesulfonic acid hydrate (MES-hydrate), pH 5.6 adjusted with KOH, 10 mM MgCl_2_, and 200 µM acetosyringone). The final bacterial OD_600_ was adjusted to 0.5. The agroinfiltration mixtures were obtained by mixing the bacteria harboring TRV1 and TRV2-PDLP/TRV2-SRC2/TRV2-eGFP constructs in a 1:1 (*v*/*v*) ratio. The TRV2-eGFP was used as the negative control, whereas the TRV2-PDS was used as a positive control for silencing. The final mixtures were kept in the dark at room temperature for 3 h before agroinoculation. For the VIGS study, the plants were categorized into four distinct groups (three biological replicates per group), TRV1 + TRV2-*NtPDLP*, TRV1 + TRV2-*NtSRC2*, TRV1 + TRV2-*eGFP* (negative control), and TRV1 + TRV2-*NbPDS* (positive control for silencing). Two matured leaves per plant were agroinoculated using a needleless syringe on the abaxial side of the leaf. Subsequently, the plants were maintained in a growth chamber, during which tissue sampling was performed at different time points (2, 4, 6, 9, 14, and 21 days post agro-inoculation), and were subjected to RNA analysis, CMV inoculation, and disease symptom observation.

### 4.5. Mechanical Inoculation of CMV in Agroinfiltrated N. benthamiana Plants

Sap inoculation of CMV in plants was performed using carborundum powder (Appendix A). The inoculating sap was prepared by collecting sap material from previously CMV-infected plants and gently rubbed on the adaxial side of the leaf using carborundum powder. For CMV inoculation in the silenced plants, leaves were initially agroinfiltrated with the VIGS constructs, and two days later, sap inoculation with CMV was performed to allow sufficient time for silencing to be initiated. Systemic leaf samples were collected at different time points (7, 12, and 19 days post inoculation (dpi) with CMV) and RNA was isolated to assess the viral titer in the silenced plants via quantitative RT-PCR (RT-qPCR).

### 4.6. PCR and RT-qPCR Analysis

PCR was performed to detect the viral titer of CMV using gene-specific primers (Table 1). The total reaction volume (20 µL) consisted of 2 µL of the 10X PCR Buffer (ThermoFisher Scientific, Waltham, MA, USA), 0.5 µL of 10 mM dNTPs, 0.5 µL of 10 μM of forward and reverse primers each (Table 1), and 0.1 µL of AmpliTaq DNA polymerase (250 U/μL) (ThermoFisher Scientific, USA). PCR was carried out with initial denaturation temperature at 95 °C for 5 min followed by 30 cycles consisting of denaturation at 95 °C for 30 s, annealing at 58 °C for 30 s, and extension at 72 °C for 30 s, and then final extension for 7 min at 72 °C.

For the validation of gene silencing, RT-qPCR was performed by extracting RNA from the systemic leaves of the silenced plants at different time points with the Plant/Fungi Total RNA Purification kit (NORGEN Biotek Corporation, Canada) according to the manufacturer’s instructions. Three biological replicates were pooled for each experimental group. The investigation of pTRV2-*NtPDLP* and pTRV2-*NtSRC2* silencing was performed by sampling tissues at 2, 4, 6, 9, 14, and 21 dpa. Tissue sampling for the detection of CMV *CP* (proxy for CMV titer) was performed at 7, 12, and 19 days post inoculation (dpi) with CMV. The RNA samples were quantified, and their purity was determined spectrophotometrically with NanoDrop™ 2000 Spectrophotometer (ThermoFisher Scientific, USA). cDNA was synthesized using 200 ng total RNA in a 20 μL reaction volume, using the SuperScript VILO cDNA synthesis kit (Invitrogen, Waltham, MA, USA) as per the manufacturer’s instructions. RNaseOUT^TM^ Recombinant Ribonuclease Inhibitor safeguards against the degradation of target RNA due to ribonuclease contamination. The 5X VILO^TM^ Reaction mix consists of random primers, MgCl_2,_ and dNTPs in a buffer formulation optimized for RT-qPCR. The RT reaction was performed in a thermocycler (BioRad, Hercules, CA, USA) at 25 °C for 10 min, followed by incubation at 42 °C for 60 min, and terminated at 85 °C for 5 min. The RT products were used as a template for amplifying the *PDLP* and *SRC2 genes* using the designed gene-specific primers (Table 1). PCR amplification for the detection of *NtPDLP*, *NtSRC2*, and *Elongation factor 1 alpha* (*EF1a*) was performed in a FastGene ULTRA Cycler (NIPPON Genetics EUROPE, Düren, Germany) using the FirePol DNA Polymerase (Solis BioDyne, Tartu, Estonia) and specific primers (Table 1). The total reaction volume consisted of 10X Buffer B (Solis BioDyne, Estonia), 10 mM dNTPs, 10 μM of forward and reverse (Table 1) primers each, 25 mM MgCl_2_ and 0.2 μL (5 U/μL) *Taq* DNA polymerase (Solis BioDyne, Estonia). PCR for the *PDLP* and *EF1a* genes was carried out with initial denaturation temperature at 95 °C for 5 min followed by 35 cycles consisting of denaturation at 95 °C for 20 s, annealing at 58 °C for 20 s, and extension at 72 °C for 1 min, and then final extension continued for 7 min at 72 °C. PCR for the *SRC2* gene was carried out with initial denaturation temperature at 95 °C for 5 min followed by 35 cycles consisting of denaturation at 95 °C for 20 s, annealing at 58 °C for 15 s and extension at 72 °C for 1 min, and then final extension continued for 7 min at 72 °C. Primer specificity was verified by melting curve analysis. To obtain a semi-quantitative estimation of the time course expression profiles of *PDLP* and *SRC2* gene CMV infection, ImageJ software (v1.54m) (https://imagej.nih.gov/ij/) was employed to measure the band intensity of the gel image. In addition, band intensities were calculated for the housekeeping gene of mock and CMV-infected plants and used as control. The values for the target genes for mock and CMV-infected plants were normalized with the housekeeping gene (*EF1a*).

qPCR was performed in a CFX96^TM^ Touch Real-Time PCR Detection System (Bio-Rad, USA) to quantify the expression levels of the target genes (pTRV2-*PDLP* and pTRV2-*SRC2*) compared to the control group (plants inoculated with pTRV2-*eGFP*) and to check the viral titer (CMV) in the silenced and control plants using SYBR Green qPCR mix (ThermoFisher Scientific, USA). Expression of the target genes was normalized with the housekeeping gene *EF1a* that was found more stable to F-box based on *BestKeeper* housekeeping gene analysis [49], agreeing with [47]. The fold change was calculated using the delta CT (2^−ΔΔCt^) method [50].

Statistical analyses were performed using Microsoft Office Excel (Microsoft Excel, Redmond, Washington, DC, USA). Student’s *t*-test was used for significance analysis between two treatments. The standard error is shown as the standard deviation between the biological replicates.

### 4.7. Histochemical Analysis

#### 4.7.1. Staining for Callose Deposition in Leaf Tissues

Aniline blue staining facilitates the detection and quantification of cell wall depositions of callose. Callose deposition assay using Aniline blue staining was performed following the procedure described in [51]. A total of three biological replicates were taken for the callose deposition test. Two infected leaf samples were harvested and transferred to 50 mL polypropylene tubes containing a destaining solution (1:3 acetic acid/ethanol). The tubes were incubated overnight with mild agitation. The destained leaves were washed with 150 mM K_2_HPO_4_ (washing solution) for 30 min. The plant materials were incubated in an aniline blue staining solution (1.3 g of K_2_HPO_4_ with 5 mg aniline blue in 50 mL H_2_O) for at least 2 h in the polypropylene tubes wrapped in aluminium foil for light protection. The samples were embedded in 50% glycerol on a glass slide and the callose deposition was visualized using a fluorescent microscope equipped with a UV lamp. Glycerol ensures a prolonged observation time and reduces bubble formation. The callose deposits appear stained bright blue in color (Appendix A).

#### 4.7.2. Staining for H_2_O_2_ Detection in Leaf Tissues

The 3,3- diaminobenzidine (DAB) assay was performed following the procedure outlined in [52]. The DAB stain is mainly used to visualize the production of H_2_O_2_ in leaves and roots. For the detection of H_2_O_2_, 0.1 g of DAB was dissolved in 100 mL of ddH_2_O (pH adjusted to 3.8 using HCl). The DAB solution was freshly prepared before starting the experiment. The DAB assay was conducted using three biological replicates for each experimental group. Systemic leaves were harvested and treated with the DAB solution and incubated overnight in the dark with mild agitation (50 rpm). The DAB solution was discarded, leaves were rinsed with ddH_2_O to remove excessive staining solution and immersed in 96% ethanol. The samples were then incubated overnight to remove the chlorophyll. Leaves were washed with 75% ethanol and kept in 20% glycerol solution to prevent the leaves from breaking easily and enabling rehydration. Dark brown color precipitates indicated accumulation of H_2_O_2_. The images of the leaf samples were taken under a light microscope with a 5X objective lens (Appendix A).

## 5. Conclusions

A better understanding of the functional roles of *PDLP* and *SRC2* in the plant’s response to CMV infection is provided. This experimental study suggests that *PDLP* and *SRC2* are potential susceptibility and resistance genes during CMV infection in *Nicotiana benthamiana*, respectively. The VIGS method can be applied for screening candidate genes involved in viral resistance or disease susceptibility in *C. chinense* cv Bhut Jolokia and other crop plants.

## Figures and Tables

**Figure 1 plants-14-00495-f001:**
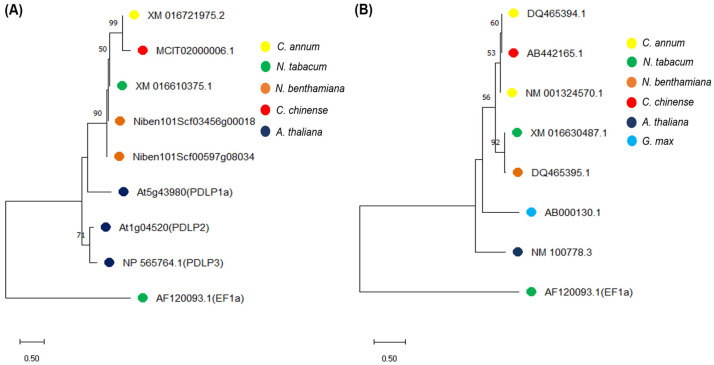
Phylogenetic analysis of (**A**) *PDLP*- and (**B**) *SRC2*-related genes from related plant species using the neighbor-joining (NJ) method. Each branch point is a representation of 10,000 bootstrap replicates. Bootstrap value higher than 50 is shown here. *Nicotiana tabacum* elongation factor 1-alpha gene sequence was used as an outgroup.

**Figure 2 plants-14-00495-f002:**
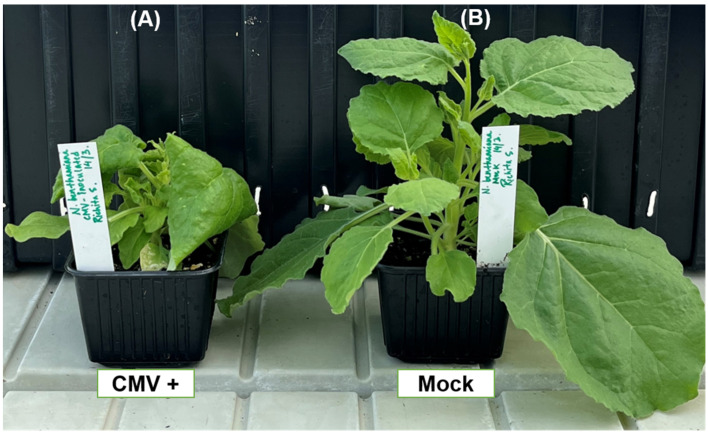
Phenotypic observation of *Nicotiana benthamiana* plants. (**A**) Cucumber mosaic virus (CMV)-infected plant and (**B**) mock (phosphate buffer)-treated plant at 14 days post treatment.

**Figure 3 plants-14-00495-f003:**
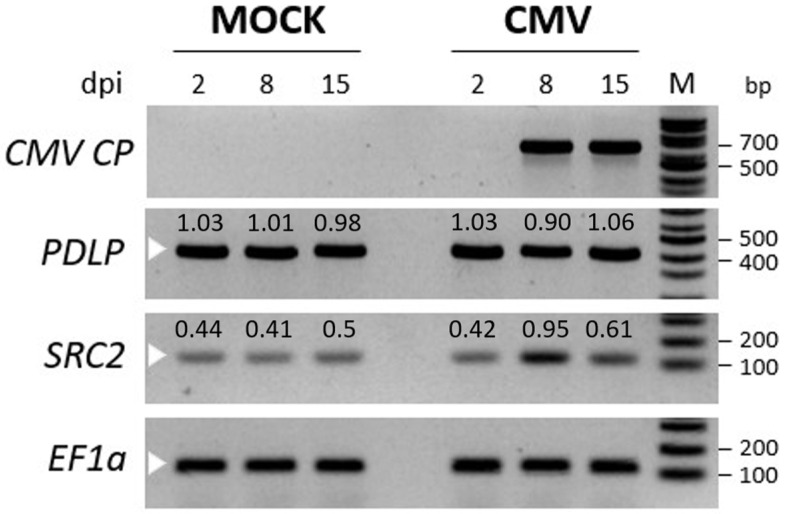
Time course expression profiles of *PDLP* and *SRC2* genes upon cucumber mosaic virus (CMV) infection in *Nicotiana benthamiana*. A semi-quantitative RT-PCR analysis is shown. A fragment of CMV coat protein gene (*CP*, 657 bp) and *N. benthamiana PDLP* (413 bp) and *SRC2* (123 bp) were amplified from cDNA obtained from systemic leaves at 2, 8, and 15 days post inoculation with CMV or phosphate buffer (mock). PCR was performed for 30 cycles and three biological replicates were pooled for each experimental group. The *elongation factor 1 alpha* (*EF1a*) gene fragment (116 bp) from *N. benthamiana* was used as an internal control. White color arrows indicate amplified target gene fragments. M indicates 100 bp ladder (New England BioLabs, NEB, Ipswich, MA, USA). The values above the bands represent the ratio of band intensity of normalized *PDLP* and *SRC2* genes in mock and CMV-infected *N. benthamiana* plants.

**Figure 4 plants-14-00495-f004:**
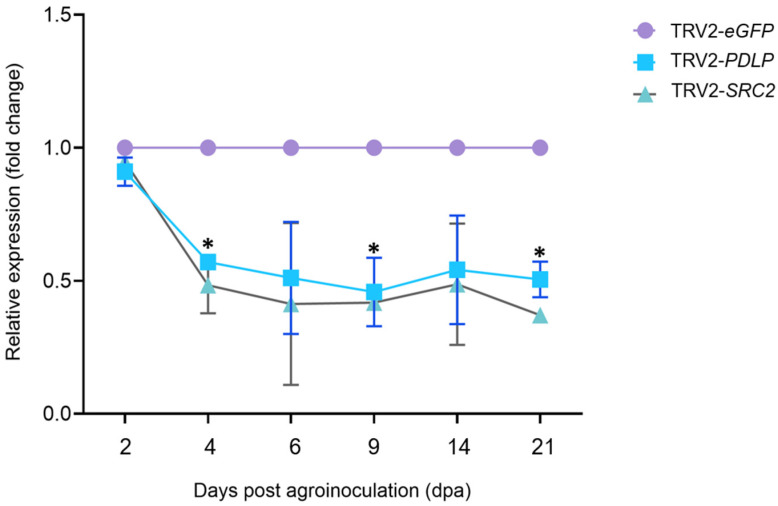
Time course expression analysis of *NtPDLP* and *NtSRC2* genes in silenced *Nicotiana benthamiana* plants. Relative quantification of *NtPDLP* and *NtSRC2* gene expression in silenced and control (TRV2-*eGFP*) plants at different time points. In addition, 50% and 60% percent downregulations of *PDLP* and *SRC2* genes were observed in *N. benthamiana* 14 days post agroinoculation. Purple color indicates control-treated (eGFP) group, blue color indicates pTRV2-*NtPDLP*-treated and green color indicates pTRV2-*NtSRC2*-treated plants. Asterisk (*) indicates significant differences between the control and silenced plants, *p*-value < 0.05 (Student’s *t*-test); dpa = days post agroinoculation.

**Figure 5 plants-14-00495-f005:**
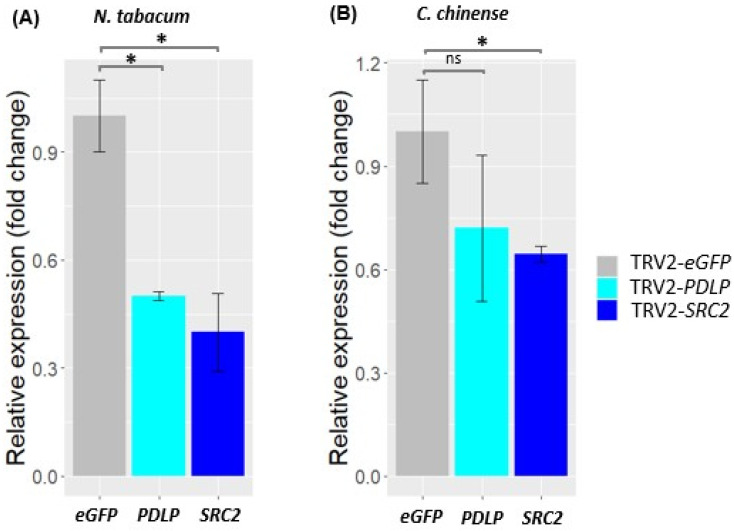
Expression analysis of *PDLP* and *SRC2* genes in silenced *Nicotiana tabacum* and *Capsicum chinense* plants at 14 days post agroinoculation (dpa). eGFP indicates the control-treated plants. Fifty and thirty percent downregulation of *PDLP* gene was observed in *N. tabacum* and *C. chinense* (Bhut Jolokia), respectively, whereas sixty and forty percent downregulation of *SRC2* gene was observed in *N. tabacum* and *C. chinense*, respectively. Student’s *t*-test was performed for mean comparisons. Asterisk (*) indicates significant difference in gene expression level between *PDLP*-, *SRC2*-silenced plants, and control (*eGFP*) plants (*p* < 0.05, Student’s *t*-test). ns: non-significant.

**Figure 6 plants-14-00495-f006:**
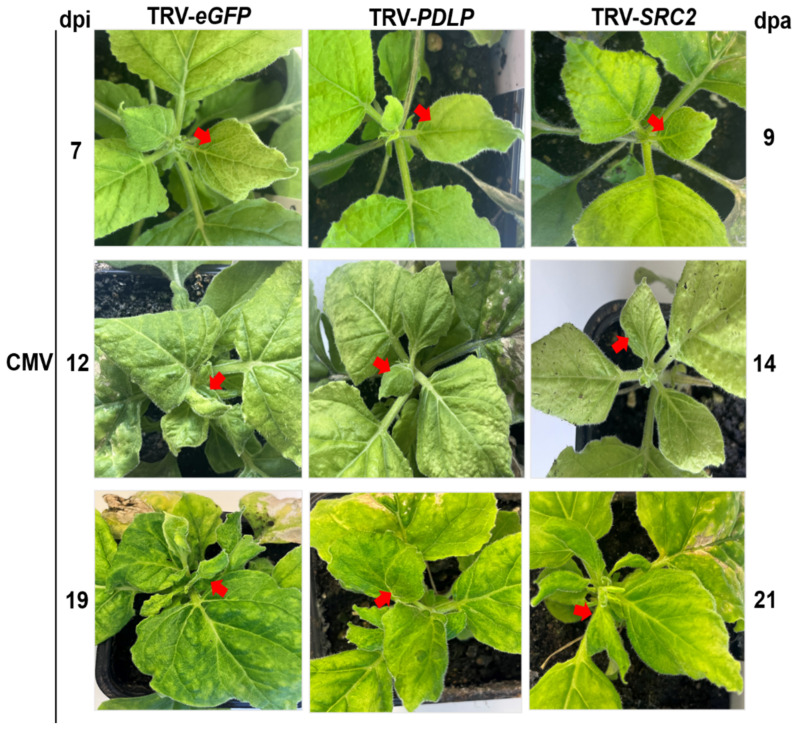
Phenotypic observations of VIGS-silenced *Nicotiana benthamiana* plants at different time-points after cucumber mosaic virus (CMV) inoculation. Plants were first agroinfiltrated with VIGS constructs, followed by sap inoculation with CMV two days later to ensure the initiation of gene silencing. In the control (*eGFP*) plants, initial symptoms of CMV infection were observed at 7 days post infection (dpi). Subsequently, symptoms such as downward leaf curling and mosaic symptoms appeared more prominently at 12 and 19 dpi. The *PDLP*-silenced *N. benthamiana* plants showed reduced viral symptoms in comparison to the control and *SRC2*-silenced plants. Red arrows indicate typical disease symptoms. dpa = days post agroinoculation.

**Figure 7 plants-14-00495-f007:**
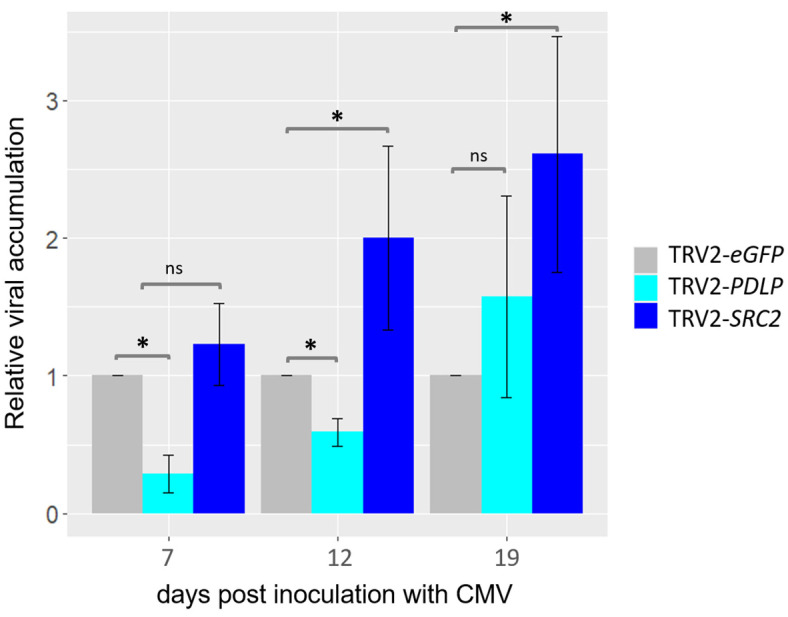
Relative abundance (RT-qPCR) of cucumber mosaic virus (CMV) titer in *PDLP*- and *SRC2*-silenced *Nicotiana benthamiana* plants at 7, 12, and 19 days post inoculation (dpi) with CMV. Plants agroinoculated with GV3101 strain carrying pTRV2-eGFP plasmids were used as control. The *elongation factor 1 alpha* (*EF1a*) gene fragment from *N. benthamiana* was used for internal control and normalization purposes. Student’s *t*-test was performed for mean comparisons. Asterisk (*) indicates significant difference in relative viral titer between *PDLP*-, *SRC2*-silenced plants, and control (*eGFP*) plants (*p* < 0.05, Student’s *t*-test). ns: non-significant.

## Data Availability

All data supporting this study are available within the paper and within the Appendix A published online.

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
