# Peer review of "Silencing of Putative Plasmodesmata-Associated Genes PDLP and SRC2 Reveals Their Differential Involvement during Plant Infection with Cucumber Mosaic Virus"

_plants, 2025, doi:10.3390/plants14030495_

Round 1
Reviewer 1 Report
Comments and Suggestions for Authors
Dear Authors
The work was done to an excellent standard. An interesting topic for plant physiology has been explored. Moreover, this article pleasantly stands out from the vast majority of works. I believe that the work corresponds to the topic of the journal, the experimental level corresponds to the level of the journal.
I only have a few small comments.
1. The introduction is written amazingly, everything is as clear as possible for a non-specialist in the field of plant virology. However, the abstract and keywords are written very unclearly. The first impression of the article is blurry. I recommend fixing it.
2. Chapter materials and methods. L500, Eppendorf termocycler? Are you sure?
The subchapter statistical analysis is missing. Briefly, what software, what statistical methods.
With best regards
Reviewer 2 Report
Comments and Suggestions for Authors
The manuscript “Silencing of Putative Plasmodesmata-Associated Genes PDLP and SRC2 Reveals their Differential Involvement during Plant Infection with Cucumber Mosaic Virus” by Saikia and colleagues aims to demonstrate the role of PDLP and SRC2 genes for CMV infection development. However, the manuscript contains insufficient data for making any conclusions and lacks a lot of controls.
Major points:
- authors didn’t show any functional activity or localization of the studied genes and proteins, respectively; they amplified sequences based on homology, however, this is not enough for claiming that the gene/protein possess similar function and localization as the known one; moreover, all sequences are put in the gene bank as putative, thus based on automatic computational analysis; for example the gene referred to as NtPDLP is designated as PREDICTED: Nicotiana tabacum cysteine-rich repeat secretory protein 3-like with predicted localization in plasmodesmata…
- VIGS is more or less standard approach in the studies of particular genes/proteins functions, thus, in my opinion, all technical procedures' description (cloning, primers, constructs verification etc) should be moved to the M&M section; moreover, I recommend authors to remove the description of this approach from the abstract (only keeping the mentioning of the method – downregulation by VIGS) and from introduction
- in studies of genes/proteins functions and roles usually both downregulation and upregulation are exploited, thus the study should be supplemented with the other approach – the target genes transient overexpression or obtainment of transgenic plants;
- PDLPs are the members of a protein family that includes proteins with different functions and different mechanism of action, thus it should be indicated, which PDLP in particular is studied
- data on callose deposition and ROS accumulation should be quantified and presented as a table or histogram with mean values and SE
- the amount of TRV RNA should be assessed for different vectors used in the study, because it could be accumulated at different levels depending of the insert or the effect on the endogenous genes
- data on fig 2 should be quantified if authors claim the difference, at least bands densitometry should be performed
Minor points:
- table with primers should be in M&M section
- improvement of the introduction:
L 39-41 – another reference should be here, maybe virus taxonomy or some review of viral movement
L42-44: authors refer to the paper by Amari et al, 2010, however, almost 15 years passed and a lot of novel information appeared; it addition, reticulons, ankyrin-repeat containing protein, A.t. beta-glucanase 2, SYTA should be mentioned here
L45: there is a paper about reticulons and CMV doi: 10.1111/mpp.13261
also about PD proteins and CMV 10.1093/jxb/erad190
L 52-53 is not correct as there are several papers on PDLPs role in viral movement, moreover, the effect of different PDLPs on PD permeability and mechanism underlying its action depends on particular member of this family
- abstract: “In contrast, SRC2-silenced plants showed enhanced susceptibility to CMV infection compared to the control (eGFP) plants” – it is not clear here why GFP is mentioned and why eGFP) plants are control
Reviewer 3 Report
Comments and Suggestions for Authors
The manuscript by Saikia et al. reports the results of the study elucidating the roles of two tobacco genes, NtPLDP and NtSRC2, in host defense response against cucumber mosaic virus (CMV) infection. The objective of this study was clear and the experiments were performed using a reverse genetic method in an appropriate manner. The results presented here are informative and intriguing. Although the manuscript looks good in general, I believe that it could be improved by paying attention to the following issues:
As for Figure 5, more detailed description of the experiment is needed. The first row of the panel is labeled with dpi 7 (left) and dpa 9 (right). Does this mean that the plants were inoculated with CMV two days after the introduction of the VIGS vector? I think that the manuscript should be described in detail so that no guesses are needed.
In the figure legend of Figure 7, the authors included a description “the intensity of the color correlates with viral load’. It is evident that the ROS accumulation was detected by DAB staining. However, it was not described how the viral load quantitation was determined.
The authors described that there would be a PDLP gene family comprising 8 genes in Arabidopsis. Then, it is likely that tobacco plant genome also contains a PDLP gene family. If there is a PDLP gene family, it will be important to know how many PDLP genes are influenced by the gene silencing. Have you checked this possibility?
The word ‘performed’ in line #157 should be replaced by ‘analyzed’.
Hyphen was either misused or missing sometimes.
Round 2
Reviewer 2 Report
Comments and Suggestions for Authors
The authors improved the manuscript and added the analysis of PDLP and SRC2 genes and proteins from different species. However, there are still questions and concerns about results and conclusions.
- the results on ROS accumulation are not convincing, I assume, that authors demonstrated a representative leaves from each group of plants, however, it is not obvious for me that there is a significant difference. If it is not possible to quantify ROS using this protocol then another one should be used (the variety of approaches is discussed for example in 10.1039/d1cb00071c). Otherwise, no conclusion based on these results should be made and the result should be moved to supplementary
- the results of callose staining are still look not convincing (at least for me), although they were moved to supplementary material. I can’t detect the difference between control and silenced plants visually, moreover, the protocol (10.21769/BioProtoc.1429.http) used for callose staining claims that callose depositions could be quantified as it was performed in the paper by the same authors (10.1105/tpc.114.126763). And for future research I could suggest the authors use another protocol for callose quantification (10.1007/978-1-4939-1523-1_7). Thus, the results on callose should not be taken into account when making the conclusions.
- I wonder, why higher callose depositions correlate with higher CMV titer according to authors point of view. This observation is mentioned in the Discussion section. “…in PDLP silenced plants, lower callose deposits and H2O2 accumulation were observed due to reduced viral accumulation…” I do not agree with that: why authors claim that callose depositions increase in response to virus reproduction? are there any references to such examples? According to my knowledge, A. thaliana PDLP1 and PDLP5 are shown to be a plasmodesmata negative regulators that enhance callose deposition thus in plants with PDLP1 knockdown the level of callose is expected to be decreased.
Similarly, the phrase (L29-30) describing causal relationships between callose/ virus, ROS/virus accumulation should be removed or modified.
- authors added the phylogenetic analysis of PDLP genes from different species as Fig 1. The readers would appreciate if the color for C. chinense and A. thaliana genes would be more different as now it is hard to distinguish between them.
- L 377-380 “This finding is consistent with the results obtained from our PDLP-silenced N. benthamiana plants, where a reduction in PDLP expression led to decreased CMV accumulation, as demonstrated by both the over-expression and silencing experiments”. What overexpression experiments are referred to here?
- authors performed densitometry of the bands presented in Fig 3, but how many replicates were performed? I thought that this figure shows the representative gel, but it seems to me now that it was the only experiment which is not enough to claim that one gene is responsive to viral infection and the other - not. In addition, if this approach is used to quantify RNA accumulation than the gradient (different cycle number) should be applied to catch the difference. There might be 30 cycles – is a plateau and analyzing the product at lower cycle number could reveal a difference between samples.
- student’s t-test – should be capitalized: Student’s t-test (Fig 5 legend)
Author Response
Comments from Reviewer 2:
NOTE: The Discussion section was re-written by placing the discussion for PDLP and SCRC2 in separate paragraphs for clearer reading.
The authors improved the manuscript and added the analysis of PDLP and SRC2 genes and proteins from different species. However, there are still questions and concerns about results and conclusions.
- the results on ROS accumulation are not convincing, I assume, that authors demonstrated a representative leaves from each group of plants, however, it is not obvious for me that there is a significant difference. If it is not possible to quantify ROS using this protocol then another one should be used (the variety of approaches is discussed for example in 10.1039/d1cb00071c). Otherwise, no conclusion based on these results should be made and the result should be moved to supplementary.
Response: We moved Fig. 8 to the supplementary material (Fig. S7) as suggested. We further rephrased the corresponding text for ROS accumulation in Results and Discussion sections.
- the results of callose staining are still look not convincing (at least for me), although they were moved to supplementary material. I can’t detect the difference between control and silenced plants visually, moreover, the protocol (10.21769/BioProtoc.1429.http) used for callose staining claims that callose depositions could be quantified as it was performed in the paper by the same authors (10.1105/tpc.114.126763). And for future research, I could suggest the authors use another protocol for callose quantification (10.1007/978-1-4939-1523-1_7). Thus, the results on callose should not be taken into account when making the conclusions.
Response: We have modified the results (lines 229-236 of the revised MS) and discussion part (Lines 325-338 of the revised MS).
As recommended by the reviewer, we plan to adopt an alternative protocol for callose and ROS quantification in future studies.
- I wonder, why higher callose depositions correlate with higher CMV titer according to authors point of view. This observation is mentioned in the Discussion section. “…in PDLP silenced plants, lower callose deposits and H2O2 accumulation were observed due to reduced viral accumulation…” I do not agree with that: why authors claim that callose depositions increase in response to virus reproduction? are there any references to such examples? According to my knowledge, A. thaliana PDLP1 and PDLP5 are shown to be a plasmodesmata negative regulators that enhance callose deposition thus in plants with PDLP1 knockdown the level of callose is expected to be decreased.
Response: We agree with reviewer’s comment. Indeed, previous studies (Caillaud et al., 2014; Lee et al., 2011) have demonstrated that overexpression (O/E) of PDLP1 and PDLP5 enhanced callose deposition at PD. We added the reference Lee et al., 2011 and a corresponding text in the Discussion. Consistent with these previous studies, our results indicated that silencing of the NtPDLP gene resulted in reduced callose deposition.
In addition, Carella et al. (2015) observed that overexpressing of either PDLP1 or PDLP5 in Arabidopsis plants resulted in a compromised SAR. We could hypothesize that silencing of PDLP in our pathosystem would augment SAR for the benefit of the plant N. benthamiana and having negative effect for the virus (CMV).
Regarding the” callose depositions increase in response to virus reproduction?” comment, the authors state that it was written by mistake and thus we deleted that sentence from the Discussion.
- Similarly, the phrase (L29-30) describing causal relationships between callose/ virus, ROS/virus accumulation should be removed or modified.
Response: We have removed the phrase (L24-26) from the Abstract.
- Authors added the phylogenetic analysis of PDLP genes from different species as Fig 1. The readers would appreciate if the color for C. chinense and A. thaliana genes would be more different as now it is hard to distinguish between them.
Response: We have changed the color for C. chinense has been changed to red color for more visibility in Fig. 1.
- L 377-380 “This finding is consistent with the results obtained from our PDLP-silenced N. benthamiana plants, where a reduction in PDLP expression led to decreased CMV accumulation, as demonstrated by both the over-expression and silencing experiments”. What overexpression experiments are referred to here?
Response: We have adjusted the text according to reviewer’s correct comment (L. 383-384).
- Authors performed densitometry of the bands presented in Fig 3, but how many replicates were performed? I thought that this figure shows the representative gel, but it seems to me now that it was the only experiment which is not enough to claim that one gene is responsive to viral infection and the other - not. In addition, if this approach is used to quantify RNA accumulation than the gradient (different cycle number) should be applied to catch the difference. There might be 30 cycles – is a plateau and analyzing the product at lower cycle number could reveal a difference between samples.
Response: Three biological replicates were performed; tissue of mock and CMV-treated plants was pooled as the material to perform RNA extraction and RT-PCR. The 30 cycles used are within the range of semi-quantitative PCR analysis. Besides, a small difference in quantity of PCR product was seen in SRC2 upon CMV infection at 8 dpi. We have added this information to the figure legend.
In addition, in the Discussion (L. 264-266) we modified the text reading now “… that the expression of PDLP is not significantly affected by CMV infection.”
- student’s t-test – should be capitalized: Student’s t-test (Fig 5 legend)
Response: We have made the change requested in Fig 5 as suggested.
Round 3
Reviewer 2 Report
Comments and Suggestions for Authors
Authors have significantly improved the manuscript. They have not removed the conclusions based on the results on callose and ROS, however, these conclusions look now more like suggestions which is much better.
Several minor points left:
- Authors claim (L 237) that “In the PDLP-silenced plants, lower H2O2 accumulation was observed (Figure S7) possibly due to reduced CMV accumulation as compared to the TRV2-eGFP-treated plants”. As the control with downregulated PLDP and without CMV infection is absent, this claim is rather speculative, thus it should be corrected (e.g. “it could be speculated that lower H2O2 accumulation was observed due to reduced CMV accumulation….. However, it could not be excluded, that the observed effect is a result of PDLP silencing” – it is just an example to illustrate my concern).
- I recommend authors to remove “PDLP proteins” and replace it with “PDLPs” or “PDLP” as the acronym includes “protein”;
- also I think that concluding paragraph (section Conclusion) should be corrected/re-written as now it is a little bit confusing from the linguistic point of view. I’m sure that “both” should be removed, “resistant” replaced by “resistance”. And I would recommend authors to refresh this phrase using the help of English native speaker.
Author Response
Comments and Suggestions for Authors
Authors have significantly improved the manuscript. They have not removed the conclusions based on the results on callose and ROS, however, these conclusions look now more like suggestions which is much better.
NOTE: We have accepted all changes made in R2 version. Changes in R3 are in track changes.
Several minor points left:
Q1. Authors claim (L 237) that “In the PDLP-silenced plants, lower H2O2 accumulation was observed (Figure S7) possibly due to reduced CMV accumulation as compared to the TRV2-eGFP-treated plants”. As the control with downregulated PLDP and without CMV infection is absent, this claim is rather speculative, thus it should be corrected (e.g. “it could be speculated that lower H2O2 accumulation was observed due to reduced CMV accumulation….. However, it could not be excluded, that the observed effect is a result of PDLP silencing” – it is just an example to illustrate my concern).
Response: We corrected the text as suggested. After the change (line number 306-313 of the revised MS), the text reads “It can be speculated that the reduced H₂O₂ accumulation observed in the PDLP-silenced plants (Figure S7) is likely a result of decreased CMV accumulation compared to the TRV2-eGFP-treated plants. It is known that in a CMV-plant compatible interaction, there is an increase in H2O2 accumulation [40], that agrees with our observations. The reduced leaf mottling and mosaic symptoms in the PDLP-silenced plants correlate nicely with the reduced CMV titer and ROS accumulation (Figure 7, Figure S7).”
Q2. I recommend authors to remove “PDLP proteins” and replace it with “PDLPs” or “PDLP” as the acronym includes “protein”;
Response: We agree with the comment and we have made the changes (used PDLPs) as mentioned in the whole text.
Q3. also I think that concluding paragraph (section Conclusion) should be corrected/re-written as now it is a little bit confusing from the linguistic point of view. I’m sure that “both” should be removed, “resistant” replaced by “resistance”. And I would recommend authors to refresh this phrase using the help of English native speaker.
Response: We have rephrased the conclusion section to avoid the confusion. Now it reads “A better understanding of the functional roles of PDLP and SRC2 in the plant’s response to CMV infection is provided. This experimental study suggests that PDLP and SRC2 are potential susceptibility and resistance genes during CMV infection in Nicotiana benthamiana, respectively”.
